# Genetic diversity and structure in two epiphytic orchids from the montane forests of southern Ecuador: The role of overcollection on *Masdevallia rosea* in comparison with the widespread *Pleurothallis lilijae*

**María Riofrío**[1]*, **Carlos Naranjo**[1], **Alberto Mendoza**[1], **David Draper**[2], **Isabel Marques**[3]

**1** Department of Biology Sciences, Universidad Técnica Particular de Loja, Loja, Ecuador, **2** CE3C - Centre for Ecology, Evolution and Environmental Changes & CHANGE - Global Change and Sustainability Institute, Lisbon, Portugal, **3** Forest Research Centre (CEF) & Associate Laboratory TERRA, Instituto Superior de Agronomia (ISA), Universidade de Lisboa, Lisbon, Portugal

* mlriofrio@utpl.edu.ec

## Abstract

Ecuador has a high diversity of orchids, but little is known about levels of genetic diversity for the great majority of species. Understanding how orchids might adapt to changes is crucial as deforestation and fragmentation of forest ecosystems threaten the survival of many epiphytic orchids that depend on other species, such as fungi and their host trees, for germination, growth, and establishment. Overcollection and the illegal trade are also major concerns for the survival of wild populations of orchids. Despite increasing awareness, effective interventions are often limited by a lack of data concerning the impacts that overexploitation might have. To fill this gap, we tested the effects of overcollection in the genetic diversity and structure of *Masdevallia rosea*, a narrow distributed epiphytic orchid historically collected in Ecuador, in comparison with the widely distributed *Pleurothallis lilijae*. Genotyping based on AFLPs showed reduced levels of diversity in wild populations but most especially in the overcollected, *M. rosea*. Overall, genetic admixture was high in *P. lilijae* segregating populations by altitude levels while fewer genetic groups were found in *M. rosea*. Genetic differentiation was low in both species. A spatial genetic structure was found in *P. lilijae* depending on altitude levels, while no spatial genetic structure was found in *M. rosea*. These results suggest different scenarios for the two species: while gene flow over long distance is possible in *P. lilijae*, the same seems to be unlikely in *M. rosea* possibly due to the low levels of individuals in the known populations. *In situ* and *ex situ* conservation strategies should be applied to protect the genetic pool in these epiphytic orchid species, and to promote the connectivity between wild populations. Adopting measures to reduce overexploitation and to understand the impacts of harvesting in wild populations are necessary to strengthen the legal trade of orchids.

**Data Availability Statement:** All relevant data are within the paper and its Supporting information files.

**Funding:** This research was supported by SENESCYT-ECUADOR and Universidad Técnica Particular de Loja". (IM): FCT—Fundação para a Ciência e a Tecnologia, I.P., Portugal through the research unit UIDB/00329/2020 (CE3C), UIDB/00239/2020 (CEF), and under the Scientific Employment Stimulus —Individual Call (CEEC Individual) —2021.01107.CEECIND/CP1689/CT0001 . The funders had no role in study design, data collection and analysis, decision to publish, or preparation of the manuscript.

**Competing interests:** The authors have declared that no competing interest exist.

## Introduction

Tropical rainforests support most of the global biodiversity, containing a large number of endemic species [1]. They play a key role in the context of climate change as they have one of the fastest carbon sequestration rates per unit land area contributing to keeping global temperatures at safe levels [2]. Yet, deforestation increased in the tropics over the half-century resulting in the loss of more than a third of all forest cover worldwide, with formerly continuous forests now existing as isolated patches, scattered across the landscape [3]. Forest fragmentation affects population viability, dispersal, and the long-term persistence of plant species [4]. Consequently, the levels of gene flow and the spatial distribution of genetic diversity, e.g., the Spatial Genetic Structure (SGS) are often affected by habitat fragmentation [5]. Understanding how species might adapt to changing environmental conditions and the mechanisms of population dynamics are therefore needed to tackle biodiversity loss and the limits of adaptation [6].

Orchids are one of the most characteristic components of the Andean flora [7–9]. Among all countries, Ecuador has a remarkable diversity as more than 20% of all vascular plants described are orchids (ca. 4200 species of orchids) [10–12]. They are widely distributed throughout Ecuadorian ecosystems but more than 60% of orchids are epiphytes occurring in montane forests and montane cloud forests [10, 13]. Despite this high diversity, orchids are affected by several threats, mainly deforestation, as they are dependent on the forest that supports them [14]. As per its size, Ecuador has the highest rate of deforestation in South America [15, 16], due to regular anthropogenic activities like land clearing for agricultural lands, mining, and new infrastructures [17]. These activities have caused the habitat destruction and fragmentation of tropical forests [18], leaving small remnants and isolated patches that affect the survival of populations and species diversity [7]. This is particularly challenging in orchids as they depend on complex interactions with mycorrhizal fungi and pollinators (besides host trees), which are also affected by forest fragmentation [7]. Orchids face other anthropogenic threats of direct impact such as the extraction of wild orchids destined for illegal trade, which has led to the local extinction of orchid populations [11]. Even if trade remains illegal in Ecuador (and elsewhere as orchids are legally protected worldwide), effects are often neglected since no studies have evaluated their impacts on the genetic diversity of wild orchid populations in the country. This is essential to propose viable conservation strategies.

This study assessed the genetic diversity and structure of two epiphytic orchids, *Pleurothallis lilijae* Foldats and *Masdevallia rosea* Lindl. occurring in different altitudinal levels and with different distributional patterns. The two species occur in the wet tropical biome but while *P. lilijae* has a wide distribution and population abundance, *Masdevallia rosea* has a restricted and narrow distribution, being historically extracted for trade in Ecuador [19]. Nevertheless, specific studies that might allow tracking the level of extirpation from wild populations are missing. Thus, as a first step to fill this gap, we measured population sizes, quantifying all adult (flowering) plants occurring in the known populations. We then consider the following questions: 1) What is the level of genetic diversity in these two species? 2) How is genetic diversity structured among populations of each species? And 3) Do they show any evidence of spatial structure? The results provide valuable information to assist future *in situ* and *ex situ* conservation actions of these two orchids.

## Materials and methods

The research permission obtained for this work were granted by the Ministry of Environment of Ecuador, code: MAE-DNB-CM-2015-0016 Ministerio del Ambiente-Ecuador.

## Study sites and sampling

The study sites were located on "Cordillera Real" in the Andes of southern Ecuador, the eastern range of the South Ecuadorian Andes [20] where six orchid populations of *P. lilijae*, (P1-P6) and four for *M. rosea* (M1-M4) occur distributed throughout the high montane evergreen forest (Fig 1). Populations are located in three different locations occurring at 2200 (San Francisco), 2800 (El Tiro), and 3000 (Cajanuma) m a.s.l. (Table 1). San Francisco has an average annual temperature of 15.1°C and a relative humidity of 96.3% and is composed of a multi-stratified evergreen forest, with a high abundance of bryophytes and epiphytes [20]. El Tiro, located at the northern limit of Podocarpus National Park, and close to the main city of Loja is crossed by the main highway from Loja to Zamora and the old highway to Zamora (not currently used). It has an average annual temperature of 9.9° C and a relative humidity of 94.5% [21]. The ecosystem is a sub-paramo of woody, shrubby, diverse vegetation, and low-stature forests [20]. Cajanuma has an average annual temperature of 9.4° C and a relative humidity of 91.1%, dominated by sclerophyllous to subsclerophyllous and lauroid forests, and with woody strata, abundant epiphytes, and mosses [20].

The two orchid species evaluated, *Pleurothallis lilijae* Foldats and *Masdevallia rosea* Lindl., are photosynthetic epiphytic perennial orchids, that belong to the tribe Epidendreae and subtribe Pleurothallidinae. Flowers in *P. lilijae* are dark purplish brown and very small (2.5 cm) with ramicauls bearing one leaf and lacking a pseudobulb [22]. In contrast, flowers in *M. rosea* are bright pink and bigger (5–8 cm), being often found in solitary inflorescences.

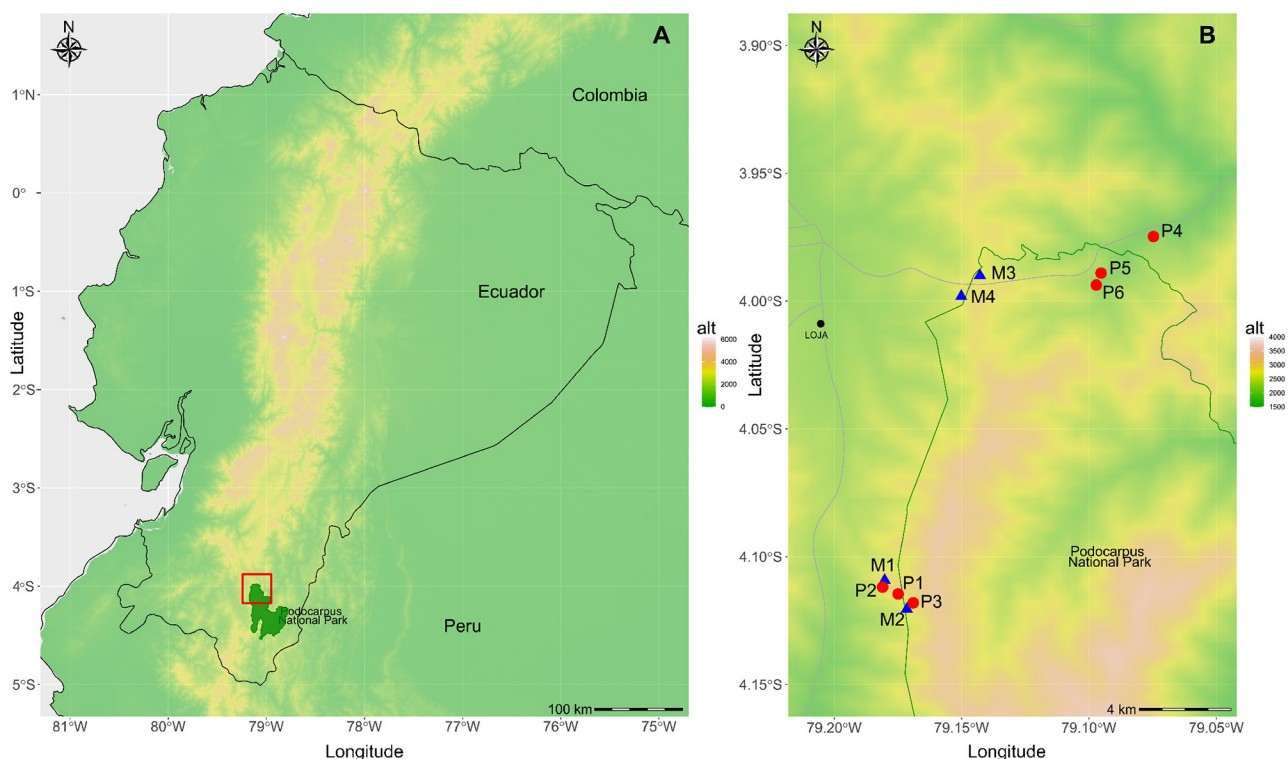

**Fig 1.** (A) Topographic map of the studied sites in Southern Ecuador. (B) Detailed location of *Pleurothallis lilijae* (red circles) and *Masdevallia rosea* (blue triangles) populations and the Podocarpus National Park, with elevation displayed. Populations IDs follow Table 1. The grey line indicates main roads. The main city (Loja) is also indicated.

**Table 1. Genetic diversity estimation for each population of two species *Pleurothallis lilijae* and *Masdevallia rosea* per population, in each location and altitudinal range.** N: number of individuals located in each population; $N_{AFLP}$: number of individuals included in genetic analysis; %P: percentage of polymorphic loci; Na: number of alleles; I: Shannon's information index; $H_e$: expected heterozygosity. Different superscript letters indicate significant differences between populations after a post hoc Tukey's test (p < 0.05).

| *Pleurothallis lilijae* | Location | Altitude | Latitude | Longitude | N | $N_{AFLP}$ | %P | Na | I | He |
|---|---|---|---|---|---|---|---|---|---|---|
| **P1** | Cajanuma | 3000 m | 702644 | 9545009 | 205 | 66 | 86.77[b] | 1.736 ± 0.019[c] | 0.280 ± 0.003[c] | 0.191 ± 0.005[c] |
| **P2** | Cajanuma | 3000 m | 702058 | 9545525 | 232 | 42 | 86.52[b] | 1.739 ± 0.019[c] | 0.290 ± 0.006[c] | 0.198 ± 0.005[c] |
| **P3** | Cajanuma | 3000 m | 703153 | 9544729 | 159 | 61 | 88.52[b] | 1.784 ± 0.018[c] | 0.270 ± 0.006[c] | 0.196 ± 0.005[c] |
| **P4** | San Francisco | 2200 m | 713993 | 9560382 | 188 | 66 | 93.80[c] | 1.878 ± 0.014[c] | 0.302 ± 0.006[c] | 0.194 ± 0.005[c] |
| **P5** | San Francisco | 2200 m | 711485 | 9558832 | 118 | 44 | 92.56[c] | 1.854 ± 0.015[c] | 0.288 ± 0.006[c] | 0.192 ± 0.004[c] |
| **P6** | San Francisco | 2200 m | 710635 | 9558497 | 116 | 49 | 87.34[b] | 1.751 ± 0.001[c] | 0.280 ± 0.006[c] | 0.198 ± 0.004[c] |
| *Overall* | | | | | *1018* | *328* | *90.55* | *1.790 ± 0.061* | *0.285 ± 0.011* | *0.194 ± 0.002* |
| *Masdevallia rosea* | | | | | | | | | | |
| **M1** | Cajanuma | 3000 m | 702043 | 9545594 | 25 | 19 | 75.30[a] | 1.208 ± 0.033[b] | 0.209 ± 0.009[b] | 0.152 ± 0.007[b] |
| **M2** | Cajanuma | 3000 m | 703171 | 9544734 | 24 | 19 | 82.23[b] | 1.249 ± 0.029[b] | 0.203 ± 0.009[b] | 0.157 ± 0.006[b] |
| **M3** | El Tiro | 2800 m | 706098 | 9558820 | 40 | 40 | 87.05[b] | 1.280 ± 0.023[b] | 0.205 ± 0.009[b] | 0.158 ± 0.006[b] |
| **M4** | El Tiro | 2800 m | 705390 | 9557895 | 18 | 17 | 73.34[a] | 1.118 ± 0.012[a] | 0.185 ± 0.009[a] | 0.109 ± 0.007[a] |
| *Overall* | | | | | *107* | *95* | *79.48* | *1.213 ± 0.070* | *0.201 ± 0.011* | *0.144 ± 0.023* |

In every orchid population, each individual plant with flowers was located, counted, and marked. For DNA extraction, we sample one orchid per phorophyte (tree host). All orchid phorophytes were georeferenced.

## Sampling for genetic analyses and DNA extraction

Young, fresh leaves were collected in adult (flowering) plants of *P. lilijae* and *M. rosea* for DNA analysis. Leaf material was dried in silica gel and stored at room temperature until DNA extraction. Total DNA was extracted from 30 mg of dried leaf material using the Plant Mini Kit (Qiagen, Hilden, Germany) following the manufacturer's instructions and stored at −20˚C. DNA concentration and quality was estimated using a fluorospectrometer (NanoDrop) and through a 1% agarose gel.

## Amplified Fragment Length Polymorphism (AFLP) amplification

AFLP reactions were performed following the procedure of [23] with the following minor modifications. Genomic DNA (approximately 500 ng) was restricted with 0.1 units of *Mse*I (New England BioLabs, Ipswich, Massachusetts, USA) and 0.5 units of *Eco*RI (Takara Bio Inc., Otsu, Japan) endonucleases, and ligated to *Mse*I and *EcoRI* adapters with 6 units of T4 DNA-ligase (Takara Bio Inc.). Samples were incubated in a thermocycler for 3 h at 37º C and 1 h at 17º C.

Preselective amplification was performed using the primers that complement the *MseI* and *EcoRI* adaptors plus one additional nucleotide i.e. *MseI*+C (5' GAT GAG TCC TGA GTA AC 3') and *EcoRI*+A (5'-GAC TGC GTA CCA ATT CA-3'). PCR reactions were conducted in 6.25 μl reactions volume containing 1.25 μl of 10-fold diluted restriction-ligation product, 1X buffer (GeneAmp 10X PCR Buffer II, Applied Biosystems, California, USA), 1.5 mM $MgCl_2$, 0.2 mM dNTPs, 0.2 μM of each primer, and 0.5 units of Taq polymerase (AmpliTaq DNA Polymerase, Applied Biosystems). The thermocycler program used for amplifications was as follows: 72 ºC for 2 min first, then 30 cycles at 94 ºC for 30 s, 56 ºC for 30 s, and 72 ºC for 2 min, with a final extension at 72 ºC for 10 min. The quality of undiluted preselective and restricted/ligation products was tested on 1% (w/v) agarose gels.

Selective amplifications were conducted in a reaction volume of 6.25 μl containing 1.25 μl of 5-fold diluted preselective product, 1X buffer (GeneAmp 10X PCR Buffer II, Applied Biosystems), 1.5 mM MgCl$_2$, 0.8 mM dNTPs, 0.08 μM of *EcoRI* fluorescent primer (Applied Biosystems), 0.2 μM of *MseI* primer (Bonsai Tech), and 0.5 U of Taq polymerase (AmpliTaq Gold DNA Polymerase, Applied Biosystems). PCR conditions were: 95 ℃ for 2 min first, 13 cycles at 94 ℃ for 30 s, 65 ℃ for 1 min (with a decreasing gradient of 0.7˚C every cycle), and 72 ℃ for 2 min; and 24 cycles at 94 ℃ for 30 s, 56 ℃ for 1 min, and 72 ℃ for 2 min, with a final extension at 72 ℃ for 10 min. Four primer combinations produced clear bands and used for selective amplification: *EcoRI* AAC NED—*MseI* CAC, *EcoRI* AGG VIC—*MseI* CTC, *EcoRI* AGA FAM—*MseI* CTA, y *EcoRI* AGA FAM—*MseI* CAC. The selection was based on preliminary screening of 10 primer combinations using six individuals from all populations. AFLP fragments were separated on an ABI 3500 sequencer (Applied Biosystems). A gene scan 500 Liz-labelled sizes standard (Applied Biosystems) was injected in each AFLP sample to obtain the size of the DNA fragments. For each species, AFLPs profiles were imported into GeneMarker v.3.0.0 (Softgenetics, LLC) and scored manually. All bands were scored as present (1) or absent (0) excluding those that could not be decisively assigned. Monomorphic and polymorphic fragments were included in the analysis. AFLP amplification was successful in 328 *P. lilijae* plants providing a total of 810 bands and in 95 *M. rosea* plants providing 664 bands. For both species, band sizes varied between 80 and 500 bp. The studied loci were 100% polymorphic.

## Detection of genetic structure

As a first step, a principal coordinate analysis (PCoA) was implemented in Genealex v. 6.5 [24] to provide a visual representation of the genetic distance relationships among individuals, considering each species. Following the suggestions of [25], a simple matching coefficient was used to calculate the distance matrix. Next, population structure was inferred by applying model-based clustering methods. Unlike the distance-based clustering, these methods use models to identify genetic groups from the full set of genotype data assigning probabilistically individuals (or a fraction of their genomes) to a specific group that represents the best fit for all patterns found. We used the approach implemented in Structure v. 2.3.4, which involves Bayesian inference and parameter estimation through Markov Chain Monte Carlo (MCMC) in the modelling process [26, 27]. We used the *admixture model* (which assumes that individuals might have a mixed ancestry) with *correlated allele frequencies* [28] and set the number of possible genetic groups (*K*) from 1 to 10. For each value of *K*, 5 independent runs were carried out using a burn-in period of 10,000 followed by 300,000 MCMC iterations. The optimal number of groups was determined by estimating the log-likelihood of the data for each *K* [ln(P (X|K)] as suggested by [26], and the ΔK statistic proposed by [29] which is based on the rate of change in the log-likelihood of data between successive *K* values. Afterward, the software CLUMPP v. 1.1.2 [30] was used, employing the *FullSearch* algorithm, to find the optimal alignment of the independent replicates, and to compute the mean membership coefficient matrix (Q-matrix). An AMOVA (analysis of molecular variance) was used to quantify the partitioning of genetic variance between and within populations with 9999 permutations at a 0.95 significance level using Genealex v. 6.5 [24].

## Detection of spatial genetic structure

The spatial genetic structure of each species was assessed using the spatial autocorrelation method proposed by [31], which allows assessing the signal generated by multiple loci. Briefly, this approach calculates an autocorrelation coefficient (*r*) between genetic and geographic

distances for all pairs of individuals within user-specified distance classes. Under isolation by distance, geographically close individuals are expected to be more genetically similar amongst themselves than to other individuals occurring at greater distances. Therefore, $r > 0$ values are expected under short-distance lags and $r < 0$ under long-distance lags. Genetic distances between individuals were calculated using the method of [32]. For distance classes, we considered a classical rule of thumb where only pairs of points separated by less than half the maximum distance observed were included [33], thus assuming a value of 9599 m for *P. lilijae* and of 7224 m for *M. rosea*. Distance classes were defined at 300 m intervals for both species when considering the entire data set. Additionally, distance classes were also established based on altitudinal range 100 m using intervals [28]. The distance between individuals was calculated using x, y, and z coordinates. Each value of *r* was tested for significant deviations from the expected value under the null hypothesis of no spatial genetic structure. This was done through a permutation test where the whole AFLP phenotypes were randomly shuffled among the occupied spatial positions, and *r*-values were recalculated each time (up to a maximum of 999 times) [31, 34]. The significance of autocorrelation (*r*) was constructed using one-tailed probability values. The significance of *r* was also tested by generating bootstrap 95% confidence intervals around the mean value of *r*. Bootstrap values were obtained by sampling, with replacement, pairs of comparisons within a given distance class [35]. Bootstrap resampling was performed 999 times and the significance of *r* was inferred when the 95% confidence interval did not contain the zero value. The significance of the entire correlogram was tested using the heterogeneity test proposed by [34]. All analyses were implemented in GenAlEx v.6.5 [24].

## Results

### Population sizes and genetic diversity

In total, 1018 flowering plants of *P. lilijae* were found in the six populations studied while, for *M. rosea* we could only locate 107 flowering plants distributed across the four studied populations (Table 1). The number of flowering plants varied significantly between populations, being overall lower for *M. rosea* than for *P. lilijae* ($F_{1,13} = 1.23$, p<0.001; Table 1).

The number of alleles was significantly higher in *P. lilijae* than in *M. rosea* populations ($F_{1,11} = 1.11$, p<0.001). This pattern was also recorded in the Shannon Diversity Index ($F = 1.04$, p<0.001), and in the observed ($F_{1,11} = 2.24$, p<0.001) and expected heterozygosity values ($F_{1,10} = 1.47$, p<0.001), being values always higher in *P. lilijae* than in *M. rosea* (Table 1). The percentage of polymorphic loci was overall very high ($F_{1,15} = 2.39$, p<0.001).

### Population genetic structure

The PCoA analysis identified two groups in *P. lilijae*, segregating populations P1, P2 and P3 occurring at 3000m from P4, P5, and P6 occurring at the lower elevation of 2200m (Fig 2A). Only 8.74% of the variance was explained by the first two axes (4.68% for PC1, and 4.06% for PC2) being the groups well divided by the PC2 axis. However, in the case of *M. rosea*, the analysis showed no clear spatial pattern (Fig 2B). Although populations M3 and M4 occurring at 2800m were generally grouped on the left of the PC1 axis, M1 and M2 occurring at 3000m were widely distributed in the two-dimensional plot (Fig 2B). The first two principal coordinates accounted for 16.63% of the total variance (11.31% PC1 and 5.32% PC2).

The Bayesian clustering analysis conducted by Structure identified four main genetic groups in *P. lilijae* based on the first value lnP(X) and the highest ΔK value, although high values were also found for $K = 6$ and $K = 8$ (S1 Fig). Populations P1, P2 and P3 where characterized by two genetic groups, distinct from the ones found in populations P4, P5, and P6 (Fig 3a), in agreement with PCoA results. For *M. rosea*, lnP(X) and ΔK results suggested the

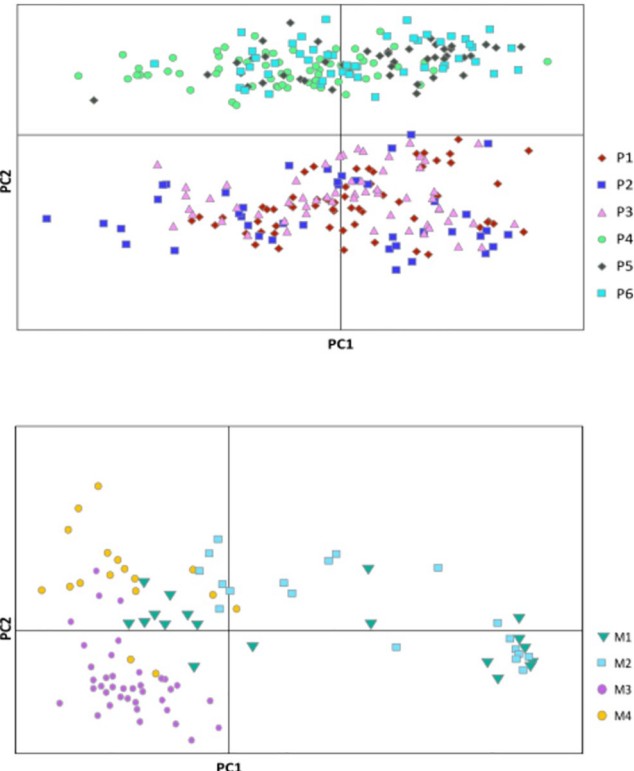

**Fig 2. Two-dimensional plot of principal coordinate analysis (PCoA) using four combinations of AFLPs and showing the clustering of (A)** *Pleurothallis lilijae* **and (B)** *Masdevallia rosea* **studied populations.**

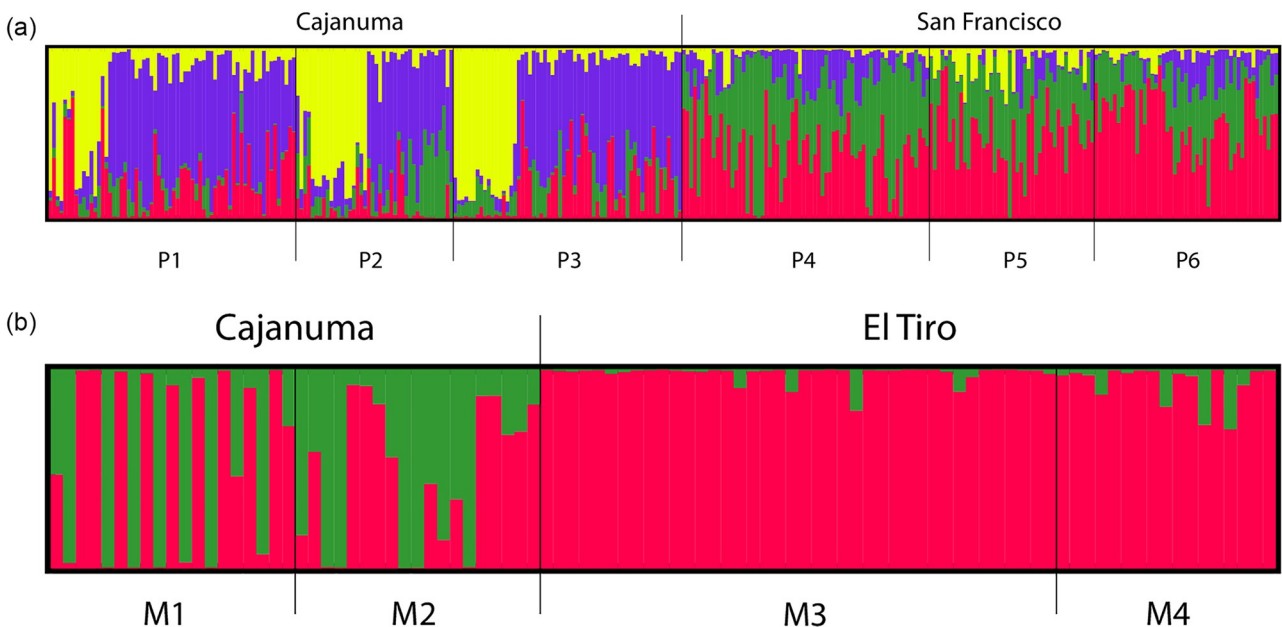

**Fig 3. Genetic population structure based on the best assignment results retrieved by STRUCTURE for K = 4 in** *Pleurothallis lilijae* **(A) and K = 2 in** *Masdevallia rosea* **(B).** Each individual plant is represented by a thin vertical line divided into K-colored segments that represent the individual's estimated membership fractions in K clusters. Names above bars indicate the location of sites (Cajanuma, San Francisco, and El Tiro). Black vertical lines delimitate the boundaries of each population. Populations IDs follow Table 1.

existence of two genetic clusters (S1 Fig) distributed throughout populations M1 to M4 (Fig 3B). In the two species, we found a high degree of genetic admixture between populations, except in M3 and M4 *M. rosea* populations where one genetic group was predominant in the analyzed plants (Fig 3).

## Population genetic differentiation

Genetic differentiation according to the analysis of molecular variance (AMOVA) indicated that 6% of all genetic variance can be attributed to differences among populations for *P. lilijae* ($\phi_{ST} = 0.057$, $P < 0.001$), while a higher level of genetic differentiation (13%) was found for *M. rosea* ($\phi_{ST} = 0.011$, $P < 0.003$). In both species, the majority of genetic variance was found within populations (Table 2).

When analyzing population differentiation by sites, the values differed (Table 2). For *P. lilijae*, molecular variance among P1-P3 populations occurring at 3000 m was 2% ($\phi_{ST} = 0.018$, $P < 0.001$), and in similar values in P4-P6 populations at 2200 m with 3% ($\phi_{ST} = 0.026$, $P < 0.001$). For *M. rosea*, molecular variance among M1-M2 populations occurring at 3000 m was 5% ($\phi_{ST} = 0.052$, $P < 0.006$), while M3-M4 populations at 2800 m showed a higher value of 12% ($\phi_{ST} = 0.119$, $P < 0.001$).

## Spatial genetic structure

When all populations were analyzed, no spatial genetic correlation was found, neither in *P. lilijae* ($\omega = 112{,}579$, $P < 0.119$; Fig 4A) nor in *M. rosea* ($\omega = 40.06$, $P < 0.447$, Fig 4B). However, when we separated data by the different sites, a positive spatial autocorrelation was observed for *P. lilijae* ($\omega = 244.169$, $P < 0.004$). A significant positive autocorrelation was obtained in the first three distance classes (100 m, $r = 0.049$, $P < 0.001$; 200m, $r = 0.031$, $P < 0.001$; 300 m, $r = 0.037$, $P < 0.001$), in distance classes between 600 up to 900 (600 m, $r = 0.03$, $P < 0.001$; 700m, $r = 0.031$, $P < 0.001$; 800 m, $r = 0.032$, $P < 0.001$; 900 m, $r = 0.028$, $P < 0.001$), 1400 to 1500 (1400 m, $r = 0.031$, $P < 0.001$; 1500 m, $r = 0.017$, $P < 0.001$), 2800 to 3000 (2800 m, $r = 0.022$, $P < 0.001$; 2900m, $r = 0.025$, $P < 0.001$; 3000 m, $r = 0.025$, $P < 0.002$), 3200 to 3400 (3200 m, $r = 0.030$, $P < 0.001$; 3300m, $r = 0.023$, $P < 0.001$; 3400 m, $r = 0.029$, $P < 0.001$), and a first x-intercept of $r$ at 2611.14m, evidencing the presence of spatial genetic structure within population in

**Table 2. Analysis of molecular variance (AMOVA) for *Pleurothallis lilijae* and *Masdevallia rosea* populations.**

| Source of variance | d.f. | Estimate variance | % Variance |
|---|---|---|---|
| *Pleurothallis lilijae* | | | |
| Among populations | 5 | 8,125 | 6 |
| Among populations within groups (P1-P3) | 2 | 2,317 | 2 |
| Among populations within groups (P4-P6) | 2 | 3,799 | 3 |
| Within populations | 322 | 134,655 | 94 |
| Within populations within groups (P1-P3) | 166 | 129,086 | 98 |
| Within populations within groups (P4-P6) | 156 | 124,484 | 97 |
| *Masdevallia rosea* | | | |
| Among populations | 3 | 10,877 | 13 |
| Among populations within groups (M1-M2) | 1 | 4,522 | 5 |
| Among populations withen groups (M3-M4) | 1 | 9,405 | 12 |
| Within populations | 91 | 74.128 | 87 |
| Within populations within groups (M1-M2) | 36 | 83,025 | 95 |
| Within populations within groups (M3-M4) | 55 | 69,74 | 88 |

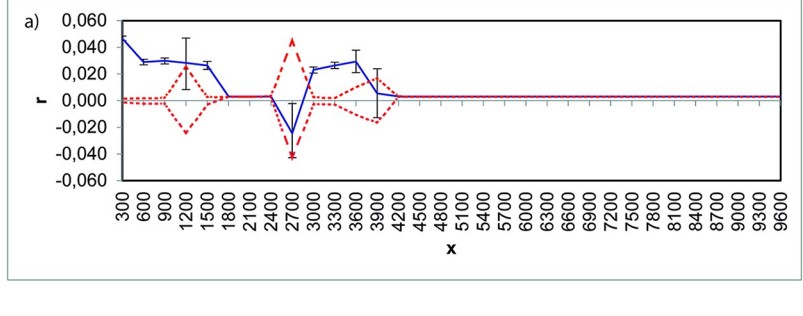

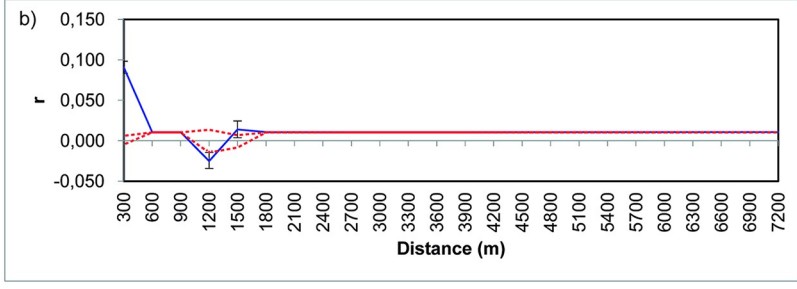

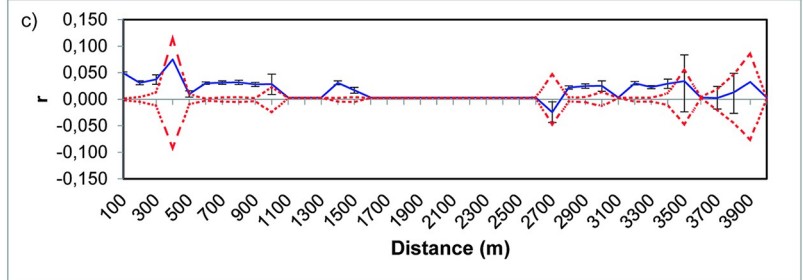

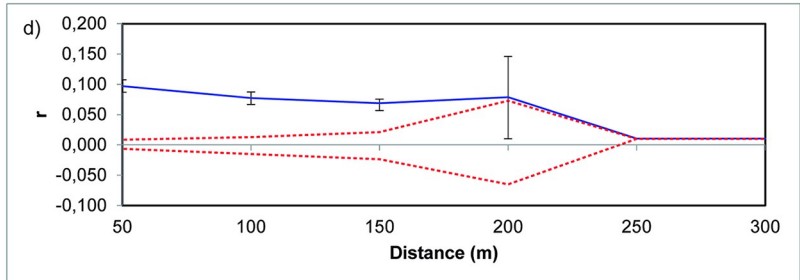

**Fig 4. Spatial genetic structure of all population of *Pleurothallis lilijae* (A) and *Masdevallia rosea* (B) and spatial genetic structure by site of *P. lilijae* (C) and *M. rosea* (D).** Dashed red lines represent upper and lower 95% confidence limits, whereas the solid blue line represents *r* (spatial autocorrelation coefficient, as a function of geographical distance).

this distance class (Fig 4C). For *M. rosea*, a significant positive autocorrelation was observed when we used the distance class according to the *intrapopulation* distance ($\omega = 49.846$, $P<0.001$). A significant positive autocorrelation was found for the first four distance classes (50 m, $r = 0.097$; $P<0.001$; 100m; $r = 0.077$, $P<0.001$; 150 m, $r = 0.069$; $P<0.001$, 200 m; $r = 0.079$, $P<0.021$) (Fig 4D).

## Discussion

This study analyzed the genetic diversity and structure of two epiphytic orchid populations occurring in three sites at different altitudes with apparent contrasting anthropogenic pressures that could affect these populations: *P. lilijae* with a large number of flowering individuals per population versus *M. rosea* that has fewer individuals due to the extraction of wild resources for trade.

### Population genetic diversity

Despite orchids are a widespread family, many species show reduced levels of genetic diversity, often promoted by habitat changes and fragmentation. For instance, the neotropical orchid, *Telipogon peruvianus*, endemic to the southern Peruvian Andes and found in only two localities shows moderate levels of genetic diversity, along with some evidence of inbreeding [36]. Likewise, the threatened *Cypripedium japonicum* [37], *Laelia rubescens* [38], *Laelia speciosa* [39], *Brassavola nodosa* [40] or *Neofinetia falcata* [41] species also show low genetic diversity values [42]. Epiphytic orchids studied here also show low patterns of genetic diversity, namely *M. rosea* (Table 1). Due to the intrinsic characteristics of orchids, such as complex symbiotic relationships with specialized pollinators and mycorrhizal fungi, orchid species are often distributed in patchy local populations [43, 44]. Anthropogenic threats such as climate change, habitat destruction, and illegal extractions further extend this pattern. Many epiphytic orchids are endangered by the high level of forest deforestation as their destiny is highly linked with their host trees [45]. This is a special concern in the Southern part of Ecuador where in the last decades, at least 20% of the forests occurring in this area were converted into pasture fields, increasing fragmentation and forest patches [46].

### Population genetic structure and differentiation

The tiny and hundreds of seeds produced within each orchid fruit offer the possibility to float in the air for long periods, facilitating long-distance dispersal [47]. As a consequence, population genetic differentiation is usually low, even across wide geographical distances or in disjunct orchid populations [48]. Based on microsatellites, $F_{ST}$ values ranged between $0.161 \pm 0.027$ in terrestrial orchids, $0.109 \pm 0.032$ in epiphytic, and $0.112 \pm 0.028$ in lithophytic orchids. Rare terrestrial orchid species show significantly higher population genetic differentiation values than common orchids (rare vs. common FST mean = 0.279 vs. FST mean = 0.092 [48]) but even so, typically less than the values reported in other plant families. Our values of genetic differentiation also agree with these general results since we found very low levels within and between populations, in the two orchid species (Table 2). The wide admixture patterns found in STRUCTURE and PCoA analysis support the existence of gene flow between populations. Pollination in orchids is usually performed by generalist insects that often lead to wide patterns of genetic admixture, even between different species of orchids [44, 49, 50]. Generalist pollination together with long-distance seed dispersal events mediated by wind [47] could have contributed to the general patterns found in our study.

Wide gene dispersion would explain the lack of spatial genetic structure when all populations were analyzed together (Fig 4), despite the presence of significant correlations at some distance classes [33]. Experimental studies often show that some orchid seeds fall within meters of the parent plant [6, 51]. Even so, in the case of *P. lilijae*, STRUCTURE and PCoA results suggest two different genetic groups segregating populations occurring at 3000 from 2200 m. Indeed, in this species, a positive spatial genetic structure was found at 2611 m, which indicates a clear boundary in gene flow [35, 52]. We have no record of pollinators for this species, but other species of *Pleurothallis* are often pollinated by small flies as *Tricimba* sp.

(Chloropidae) and *Megaselia* sp. (Phoridae) [53, 54], where flying distances between conspecific populations ranged from 13 to 587 km, and distances between different *Pleurothallis* species were recorded up to 1167 km. Dipteran families such as Sciaridae and Drosophilidae have been also reported to pollinate *Pleurothallis* species [53–56]. Thus, the action of these pollinators could explain some local patterns found here for *Pleurothallis*. This contrast with *Masdevallia* species where different floral visitors have been recorded such as bees, and hummingbirds [57], as well as flies such as *Zygothrica* [58], which could explain the different spatial genetic patterns found in the two species.

### Is genetic diversity and structure affected by overcollection due to illegal or unregulated trade?

Understanding the effects of illegal extractions from wild orchid populations is a complex subject. Despite this is a well-known conservation issue, monitoring of populations and reports of wild trade are often missing in orchids, preventing a full assessment of the impacts of extractions, either from a domestic perspective or considering the significance that the international wildlife trade might have [59]. Orchids offer a significant income to livelihoods in many low-income countries and the sustainability of this chain can be undermined by illegal trade [60, 61]. An important first step to addressing the unsustainable plant trade is to recognize that it might be a conservation issue and establish long-term monitoring studies. From our field observations, it was clear that overcollection was intensified in populations where there was easy access such as the ones in El Tiro, very close to city roads. Overcollection is prevalent on orchids that bear showy bright flowers such as *M. rosea*, among others (*pers. obs*). Here, we provided evidence that genetic diversity and structure are low in populations affected by the extraction of orchids contributing to the homogenization of the genetic pool. For instance, one single genotype group was prevalent in M3 and M4 *M. rosea* populations occurring in El Tiro (Fig 3b) and the levels of genetic diversity were lower in these populations than in the remaining ones (Table 1). In contrast, more genetic diversity and variability was found in *P. lilijae*. This, together with changes in the habitat might disrupt the long-term sustainability of natural populations of *M. rosea*, as reported in other orchids. For instance, a study based on microsatellite markers on the orchid *Pelatantheria scolopendrifolia* (Makino) has revealed a low level of genetic diversity (Ho = 0.529, He = 0.356) and high levels of population differentiation as a consequence of population decline due to illegal collection and habitat disturbances [62]. Another study based on microsatellite markers in *Cypripedium calceolus*, a rare orchid that also shows a marked decline due to habitat loss, fragmentation, and over-collection, revealed high rates of differentiation between populations [5]. Genetic connectivity between remnant patches also evidence that gene flow is often too low to maintain sustainable populations [5]. Surprisingly, high levels of genetic diversity have been also recorded in wild populations heavily used in trade such as in *Laelia speciosa*, although recent bottlenecks [63] already evidence the loss of allelic diversity.

Actions for the conservation of *M. rosea* should focus on enhancing the levels of gene flow between populations and reducing the extraction of plants. Sensibilization actions targeting local villages and stakeholders of the orchid supply chain are needed to conserve wild orchids, and the involvement of these stakeholders could be key to developing strategies for sustainable trade. New measures or legal regulations by the corresponding government entities could also help to protect the populations *in situ* and avoid illegal trade. On the other hand, *in situ* and *ex situ* conservation strategies should be considered for both *P. lilijae* and *M. rosea*. *In situ* conservation allows to maintain the genetic pool of populations while assuring the conservation of their habitat [64], and the underlying ecological networks that orchids have with other

organisms such as mycorrhizal fungi [43] or pollinators [65, 66]. *Ex situ* conservation strategies that allow the storage of seeds from all populations would assist future reintroduction plans, increase population sizes, and establish new populations to enhance gene flow and connectivity between populations. Finally, given the predictions of ongoing climate change, we recommend the establishment of long-term monitoring plots for improving knowledge of demographic data while also offering effective results to build resilient populations.

## Supporting information

**S1 Fig. Number of genetic clusters in the studied population of *Pleurothallis lilijae* and *Masdevallia rosea* according to structure.** The log-likelihood of the data [lnP(X)] averaged over 10 consecutive Structure runs for K = 1 to 10, with error bars representing ± standard deviation for *P. lilijae* (a) and *M. rosea* (c). Evanno's ΔK statistic plotted against K for *P. lilijae* (b), and *M. rosea* (d).
(DOCX)

## Acknowledgments

We thank Daniela Arias for her help in the laboratory, and M. Naranjo and A. Poma for their help in part of the data analysis.

## Author Contributions

**Conceptualization:** María Riofrío, Carlos Naranjo.

**Data curation:** Alberto Mendoza.

**Formal analysis:** María Riofrío, Carlos Naranjo, Alberto Mendoza, David Draper, Isabel Marques.

**Investigation:** María Riofrío.

**Methodology:** María Riofrío, Carlos Naranjo, Alberto Mendoza.

**Project administration:** María Riofrío, Carlos Naranjo.

**Validation:** David Draper, Isabel Marques.

**Visualization:** David Draper, Isabel Marques.

**Writing – original draft:** María Riofrío, Carlos Naranjo, David Draper, Isabel Marques.

**Writing – review & editing:** María Riofrío, David Draper, Isabel Marques.

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
