## [Decision Letter · Decision Letter 0]

14 Dec 2022

PONE-D-22-26701Understanding the genetic structure of two epiphytic orchids Pleurothallis lilijae and Masdevallia rosea in the montane forest, in southern EcuadorPLOS ONE

Dear Dr. Riofrío Guamán,

Thank you for submitting your manuscript to PLOS ONE. After careful consideration, we feel that it has merit but does not fully meet PLOS ONE’s publication criteria as it currently stands. Therefore, we invite you to submit a revised version of the manuscript that addresses the points raised during the review process. ==============================

ACADEMIC EDITOR:The work reported in the manuscript is important, however the manuscript cannot be considered for publication in its present form, hence, authors are advised to revise the manuscript as per reviewers' suggestions to improve its quality.

We look forward to receiving your revised manuscript.

Kind regards,

Vikas Sharma, Ph.D

Academic Editor

PLOS ONE

Journal Requirements: 

"This research was supported by the SENESCYT-ECUADOR and Universidad Técnica Particular de Loja. We thank Daniela Arias for her help in the laboratory, and M. Naranjo and A. Poma for their help in part of the data analysis"

"This research was supported by SENESCYT-ECUADOR and Universidad Técnica Particular de Loja"

1. You may seek permission from the original copyright holder of Figure (1) to publish the content specifically under the CC BY 4.0 license.  

Reviewers' comments:

Reviewer's Responses to Questions

**Comments to the Author**

1. Is the manuscript technically sound, and do the data support the conclusions?

Reviewer #1: Partly

Reviewer #2: Yes

2. Has the statistical analysis been performed appropriately and rigorously? 

Reviewer #1: No

Reviewer #2: Yes

3. Have the authors made all data underlying the findings in their manuscript fully available?

Reviewer #1: Yes

Reviewer #2: Yes

4. Is the manuscript presented in an intelligible fashion and written in standard English?

Reviewer #1: No

Reviewer #2: No

5. Review Comments to the Author

Reviewer #1: • In general, there are a lot of typographical and grammatical errors. Moreover authors should apply the right punctuation where necessary to enhance the understanding of the results.

• There are a lot of typographical and grammatical errors. Moreover authors should apply the right punctuation where necessary to enhance the understanding of the results.

• The analysis is not clear and robust enough

• The results and discussion are not extensive enough to reveal their genetic structure.

• Authors should provide the AMOVA results/table showing the % of variation among and within the populations

• Presenting the variance of the first 2 components under the PCoA analysis alone is not enough to discriminate the populations. The other components maybe significant as well.

Reviewer #2: This is an important study understanding the genetic diversity of orchids species Pleurothallis lilijae and Masdevallia rosea distributed in Andes of southern Ecuador. Author have not described the use and importance of these two orchids due to which these were illegally trade. Author have not described that why these two species of orchids are ecologically important to these mountains.

I have some queries regarding this manuscript as

Why there is much difference in the sample size taken for analysis in both two species?

In discussion portion, Seed dispersal method should be explained properly to understand the distribution of genetic groups of the population.

There are some sequential gaps in this manuscript which stops the reader to understand the manuscript properly. Improvement should be made in writing and filling the story gaps with valid interpretation with results to make this paper more interesting. I hope the Authors will take the above suggestion seriously and improve the manuscript.

Some of the mistakes and queries which should be taken care of are

1. Line 68 annual mean temperature is not provided for this site

2. Line 78 Species spelling mistake

3. Line 99 Not clear from the topic how many samples for each species taken for AFLP

4. Line 186 lilijae in italics

5. In Bidimensional plot of PCoA for Masdevallia rosea POP 4 Population seems to be different from POP1 and POP2

6. What are the anthropogenic factors due to which these species are threatened and should be conserved in-situ and ex-situ

6. PLOS authors have the option to publish the peer review history of their article (what does this mean?). If published, this will include your full peer review and any attached files.

Reviewer #1: No

Reviewer #2: **Yes: **Dr. Vikrant Jaryan

---

## [Author Response · Author response to Decision Letter 0]

19 Jun 2023

Dear Editor:

Thank you for the opportunity for submitting a new, improved version, in which we have incorporated all the comments raised by the reviewers. We deeply thank the reviewers for the time he/she took for performing the revision of our paper, which has led to major changes. We have re-written the manuscript (including the abstract and the title) to overcome the concerns raised, while also eliminating previous typos and mistakes that were left in the previous version. Legends of figures and tables have also been revised. Please find below (in blue in response_to_reviewers document) a point-by-point answer to the questions raised, and how we have addressed them in this new version of the manuscript highlighted with track changes. 

Answer to Journal requirements:

Authors' answer: We have checked that the paper meets all journal style requirements.

2. We note that you have provided funding information that is not currently declared in your Funding Statement. However, funding information should not appear in the Acknowledgments section or other areas of your manuscript. We will only publish funding information present in the Funding Statement section of the online submission form. Please remove any funding-related text from the manuscript and let us know how you would like to update your Funding Statement. Currently, your Funding Statement reads as follows: "This research was supported by SENESCYT-ECUADOR and Universidad Técnica Particular de Loja" Please include your amended statements within your cover letter; we will change the online submission form on your behalf.

Authors' answer: The funding statement has been updated as requested and was include in the new cover letter following instructions.

3. We note that Figure 1 in your submission contain [map/satellite] images which may be copyrighted.

Authors' answer: Figure 1 has been updated and is now a house-made map. We can make further changes if PlosOne still considers it to be necessary.

Answer to Reviewers' comments: 

1. Is the manuscript technically sound, and do the data support the conclusions? The manuscript must describe a technically sound piece of scientific research with data that supports the conclusions. Experiments must have been conducted rigorously, with appropriate controls, replication, and sample sizes. The conclusions must be drawn appropriately based on the data presented. 

Reviewer #1: Partly 

Reviewer #2: Yes

Authors' answer: In this new version, we have made a significant effort to eliminate the concerns raised. We have re-written several sections to clarify our message, while eliminating typos and incorrect sentences that were clearly disrupting the manuscript flow. We deeply acknowledge the comments and time of the reviewers, which helped us to provide a better version of our study. 

2. Has the statistical analysis been performed appropriately and rigorously? 

Reviewer #1: No 

Reviewer #2: Yes

Authors' answer: We are not sure about the concern raised by Reviewer 1, but based on the comments he/she stated below, we believe they are related to the AMOVA results, which has been stated in this new version. The text has also been re-written to better explain the results found.

3. Have the authors made all data underlying the findings in their manuscript fully available? The PLOS Data policy requires authors to make all data underlying the findings described in their manuscript fully available without restriction, with rare exception (please refer to the Data Availability Statement in the manuscript PDF file). The data should be provided as part of the manuscript or its supporting information, or deposited to a public repository. For example, in addition to summary statistics, the data points behind means, medians and variance measures should be available. If there are restrictions on publicly sharing data—e.g. participant privacy or use of data from a third party—those must be specified. 

Reviewer #1: Yes 

Reviewer #2: Yes

Authors' answer: Thank you for this positive feedback.

4. Is the manuscript presented in an intelligible fashion and written in standard English? PLOS ONE does not copyedit accepted manuscripts, so the language in submitted articles must be clear, correct, and unambiguous. Any typographical or grammatical errors should be corrected at revision, so please note any specific errors here. 

Reviewer #1: No 

Reviewer #2: No

Authors' answer: We do agree that this was a major limitation in the previous version. To accomplish our project deadlines, we have submitted the manuscript without a throughout proofreading step – we have done so in this new version. We hope that the message is now clear. Thank you for this new opportunity.

5. Review Comments to the Author Please use the space provided to explain your answers to the questions above. You may also include additional comments for the author, including concerns about dual publication, research ethics, or publication ethics. (Please upload your review as an attachment if it exceeds 20,000 characters)

Reviewer 1: 

In general, there are a lot of typographical and grammatical errors. Moreover authors should apply the right punctuation where necessary to enhance the understanding of the results. 

Authors' answer: We agree. We have carefully revised the text to eliminate this concern.

The analysis is not clear and robust enough 

Authors' answer: We do not follow the reviewer's comment. Our analytic procedure is the same one employed in many other genetic studies. We can make further changes if the reviewer specifies the reason for concern. However, we believe this concern was due to the fact that we have not explained some of the results found. In this new version, we have made a significant effort to clarify the number of individuals, populations, and sites sampled to better explain the importance of our study.

The results and discussion are not extensive enough to reveal their genetic structure. 

Authors' answer: Our results show a wide degree of genetic admixture between populations, which is common in many genetic studies devoted to neotropical orchids. Nevertheless, in one of the species, two main genetic clusters were found, clearly segregating populations by altitude. We have re-written the results and the discussion to better explain this message while comparing our results with other studies in neotropical forests and orchid species.

Authors should provide the AMOVA results/table showing the % of variation among and within the populations 

Authors' answer: We have done so in this new version.

Presenting the variance of the first 2 components under the PCoA analysis alone is not enough to discriminate the populations. The other components maybe significant as well.

Authors' answer: Please note that there is a high degree of gene flow between populations as revealed by STRUCTURE. That is congruent with the lack of spatial isolation in the PCoA results, and agrees with the general findings in other orchid species. We have re-written the results and the discussion to clarify this outcome.

Reviewer #2: 

This is an important study understanding the genetic diversity of orchids species Pleurothallis lilijae and Masdevallia rosea distributed in Andes of southern Ecuador. Author have not described the use and importance of these two orchids due to which these were illegally trade. Author have not described that why these two species of orchids are ecologically important to these mountains. I have some queries regarding this manuscript as:

Why there is much difference in the sample size taken for analysis in both two species?

Authors' answer: the sampling size reflects the number of adult plants quantified in each population. We have clarified that in this new version, including the sampling size in material and methods, and results (including the new Table 1). 

We do agree that we did not explained the differences or the importance of these orchids, and the influence of overcollection. We have re-written several sections to better explain this.

In discussion portion, Seed dispersal method should be explained properly to understand the distribution of genetic groups of the population. There are some sequential gaps in this manuscript which stops the reader to understand the manuscript properly. Improvement should be made in writing and filling the story gaps with valid interpretation with results to make this paper more interesting. | hope the Authors will take the above suggestion seriously and improve the manuscript.

Authors' answer: We do agree that the previous version had major limitations since we didn’t carefully proofread it. We have done so in this new version, re-writing several sections, eliminating typos and incorrect sentences that were disrupting our message. We have also explained the influence of gene flow, including seed dispersal, in the results obtained. Thank you for the opportunity to do so.

Some of the mistakes and queries which should be taken care of are:

1.Line 68 annual mean temperature is not provided for this site

Authors' answer: We have re-written this sentence to clarify the sites and the populations sampled.

2. Line 78 Species spelling mistake

Authors' answer: Spelling has been revised throughout the text.

3. Line 99 Not clear from the topic how many samples for each species taken for AFLP

Authors' answer: Yes, the reviewer is right. We have clarified that in this new version. Sampling is also indicated in the new table 1.

4. Line 186 lilijae in italics

Authors' answer: Thank you. This has been corrected throughout the manuscript.

In Bidimensional plot of PCoA for Masdevallia rosea POP 4 Population seems to be different from POP1 and POP2

Authors' answer: We have clarified and discussed those results in this new version. We have edited the labels of figures to better indicate the sites that were sampled.

What are the anthropogenic factors due to which these species are threatened and should be conserved in-situ and ex-situ

Authors' answer: Both orchids are threatened due to habitat fragmentation and land use changes that we have explained in this new version. Masdevallia rosea is also widely collected from wild populations. 

Answers to the Annotated pdf:

Line 95- L95. Check Qiagen

Authors' answer: Thank you; this has been corrected.

L97. is it electrophoresis period? If so please add unit eg mins, sec

Authors' answer: This has a mistake; it has been corrected.

L100. Space between “[18]with”

Authors' answer: Corrected.

L170. in “the” whole analyses

Authors' answer: This section has been re-written to clarify the text.

L171. Check “100m.[28]Distance”

Authors' answer: Corrected. 

L176-177. Please just number reference it “[see Smouse and Peakall (1999) and Smouse et al. (2008) for details].”

Authors' answer: Corrected. 

L192: reword. The studied loci were 100% polymorphic.

Authors' answer: This has been changed as requested. However, please note that this sentence has been moved to clarify our message. 

L195: should be (Fig 2A)

Authors' answer: This has been corrected throughout the paper.

L195: I see the variance of the first two components alone not enough to discirminate the population. the other components maybe significant as well

Authors' answer: Note that there is a wide admixture between populations. That is congruent with the lack of spatial clustering in a PCoA analysis and the low level of genetic differentiation. This is congruent with the results found in other orchids as explained in this new version. However, in one species we were able to separate genetic groups by altitude. We have re-written the results to explain this better.

L198: Fig 2B

Authors' answer: This has been corrected throughout the paper.

L206: Fullstop.

Authors' answer: This section has been re-written.

L209: full stop and comma

Authors' answer: This section has been re-written.

L212: Fig 4A 

Authors' answer: This has been corrected throughout the paper.

please insert Paragraph

Authors' answer: This section has been re-written.

Fig 3B&3C

Authors' answer: This has been corrected throughout the paper.

remove "see that”

Authors' answer: This section has been re-written.

L214 and 216: 0.8 not 0,8

Authors' answer: Corrected.

L216: Fig 4B

Authors' answer: This has been corrected throughout the paper.

L229: where is the AMOVA result/table showing the % of variation among and within populations

Authors' answer: that was in the text. We have include a table as requested.

L233: check "s"

Authors' answer: Corrected.

L233: this table should not be a supplementary

Authors' answer: Ok. It is now Table 1 and a new section has been added.

L234: analyzing the

Authors' answer: This section has been re-written.

L235: comma

Authors' answer: This section has been re-written.

L236: Fullstop

Authors' answer: This section has been re-written.

L240: which analysis, be specific. Punctuate (fullstop) followed by No.

Authors' answer: This section has been re-written.

L242: reword this part.

Authors' answer: This section has been re-written.

L253: All fig should be Fig

Authors' answer: This has been corrected throughout the paper.

L257: rephrase the figures description.

Authors' answer: All legends have been revised.

L269: punctuate this paragraph to enhance the understanding of the message.

Authors' answer: This has been revised.

Please discuss how the low of genetic differentiation observed could influence conservation of these species

Authors' answer: This has been included.

L280: please apply the right punctuation where necessary to enhance the undersatnding of your message

Authors' answer: This has been revised.

L287: apply the necessary punctuations

Authors' answer: This has been corrected throughout the paper.

L288: full stop

Authors' answer: This section has been re-written.

L301: from our study/research

Authors' answer: This section has been re-written.

Fullstop. Therefore, measures for the protection and conservation of this species and other orchid species has to be intensified.......

Authors' answer: This section has been re-written.

---

## [Decision Letter · Decision Letter 1]

14 Aug 2023

Genetic diversity and structure in two epiphytic orchids from the montane forests of southern Ecuador: the role of overcollection on Masdevallia rosea in comparison with the widespread Pleurothallis lilijae

PONE-D-22-26701R1

Dear Dr. Riofrío Guamán,

We’re pleased to inform you that your manuscript has been judged scientifically suitable for publication and will be formally accepted for publication once it meets all outstanding technical requirements.

Kind regards,

Mehdi Rahimi, Ph.D.

Academic Editor

PLOS ONE

Additional Editor Comments (optional):

Reviewers' comments:

Reviewer's Responses to Questions

**Comments to the Author**

1. If the authors have adequately addressed your comments raised in a previous round of review and you feel that this manuscript is now acceptable for publication, you may indicate that here to bypass the “Comments to the Author” section, enter your conflict of interest statement in the “Confidential to Editor” section, and submit your "Accept" recommendation.

Reviewer #1: All comments have been addressed

Reviewer #2: All comments have been addressed

2. Is the manuscript technically sound, and do the data support the conclusions?

Reviewer #1: Yes

Reviewer #2: Yes

3. Has the statistical analysis been performed appropriately and rigorously? 

Reviewer #1: Yes

Reviewer #2: Yes

4. Have the authors made all data underlying the findings in their manuscript fully available?

Reviewer #1: Yes

Reviewer #2: Yes

5. Is the manuscript presented in an intelligible fashion and written in standard English?

Reviewer #1: Yes

Reviewer #2: Yes

6. Review Comments to the Author

Reviewer #1: Authors have adequately addressed all my comments raised in the previous review. I therefore recommend the manuscript acceptable for publication

Reviewer #2: The manuscript is highly improved and author has gone through all the queries raised by the reviewer. I have found that the English in the manuscript is improved and now the manuscript has come in the sequence of order to understand the study. But author has not described the type of pollination occurred in these orchids? Author is advised to write the conclusion separately so that reader can understand the outcome of the study. The plant species should be identified properly and should have authenticated from some recognized herbarium. Despite these, I found there are some mistakes which should be taken care of by the author are as following

Line no. 61-63.

For its size, Ecuador has the highest rate of deforestation in South America [15,16], as a consequence of land clearing for agricultural lands, mining, and new infrastructures [17].

Reframe this sentence.. As per its size, Ecuador has the highest rate of deforestation in South America due to regular anthropogenic activities like land clearing for agriculture lands, mining, and development of new infrastructure.

Line 63

These activities have caused the habitat destruction and fragmentation of the tropical forest

Line 93-94

P. lilijae in italics, M. rosea in italics

Line 130-131.

AFLP reactions were performed following the procedure of [23] with the following minor modifications

Sentence is incomplete. Procedure of what

Line 170-171.

Following the suggestions of [25], a simple matching coefficient was used to calculate the distance matrix.

Sentence is incomplete

Line 192-193

The spatial genetic structure of each species was assessed using the spatial autocorrelation method proposed by [31]

The sentence is incomplete

Line 199-200

Genetic distances between individuals were calculated using the method of [32]

Sentence is incomplete

Line 205

100 m

Line 205-209

The sentence is not clear. Reframe the sentence. You can split the sentence into two.

Line 320-322

Likewise, the threatened Cypripedium japonicum [37], Laelia rubescens [38], Laelia speciosa [39], Brassavola nodosa [40] or Neofinetia falcata [41] species also showed low genetic diversity values [42]

7. PLOS authors have the option to publish the peer review history of their article (what does this mean?). If published, this will include your full peer review and any attached files.

Reviewer #1: No

Reviewer #2: **Yes: **Dr. Vikrant Jaryan, Associate Professor, Sant Baba Bhag Singh University, Jalandhar

---

## [Editor Report · Acceptance letter]

21 Aug 2023

PONE-D-22-26701R1 

Genetic diversity and structure in two epiphytic orchids from the montane forests of southern Ecuador: the role of overcollection on *Masdevallia rosea* in comparison with the widespread *Pleurothallis lilijae*

Dear Dr. Riofrío Guamán:

I'm pleased to inform you that your manuscript has been deemed suitable for publication in PLOS ONE. Congratulations! Your manuscript is now with our production department. 

Kind regards, 

on behalf of

Associate Prof. Mehdi Rahimi 

Academic Editor

PLOS ONE